# Day-to-Day Variability in Measurements of Respiration Using Bioimpedance from a Non-Standard Location

**DOI:** 10.3390/s24144612

**Published:** 2024-07-16

**Authors:** Krittika Goyal, Dishant Shah, Steven W. Day

**Affiliations:** 1Department of Manufacturing and Mechanical Engineering Technology, Rochester Institute of Technology, Rochester, NY 14623, USA; krgmet@rit.edu (K.G.); ds9091@rit.edu (D.S.); 2Department of Biomedical Engineering, Rochester Institute of Technology, Rochester, NY 14623, USA

**Keywords:** bioimpedance-based sensing, wearable sensors, home care follow-up, respiration monitoring, dry electrodes

## Abstract

Non-invasive monitoring of pulmonary health may be useful for tracking several conditions such as COVID-19 recovery and the progression of pulmonary edema. Some proposed methods use impedance-based technologies to non-invasively measure the thorax impedance as a function of respiration but face challenges that limit the feasibility, accuracy, and practicality of tracking daily changes. In our prior work, we demonstrated a novel approach to monitor respiration by measuring changes in impedance from the back of the thigh. We reported the concept of using thigh–thigh bioimpedance measurements for measuring the respiration rate and demonstrated a linear relationship between the thigh–thigh bioimpedance and lung tidal volume. Here, we investigate the variability in thigh–thigh impedance measurements to further understand the feasibility of the technique for detecting a change in the respiratory status due to disease onset or recovery if used for long-term in-home monitoring. Multiple within-session and day-to-day impedance measurements were collected at 80 kHz using dry electrodes (thigh) and wet electrodes (thorax) across the five healthy subjects, along with simultaneous gold standard spirometer measurements for three consecutive days. The peak–peak bioimpedance measurements were found to be highly correlated (0.94 ± 0.03 for dry electrodes across thigh; 0.92 ± 0.07 for wet electrodes across thorax) with the peak–peak spirometer tidal volume. The data across five subjects indicate that the day-to-day variability in the relationship between impedance and volume for thigh–thigh measurements is smaller (average of 14%) than for the thorax (40%). However, it is affected by food and water and might limit the accuracy of the respiratory tidal volume.

## 1. Introduction

Non-invasive monitoring of pulmonary health may be useful for tracking several health conditions, ranging from asthma to recovery from infectious disease, such as COVID-19 [1,2] and Respiratory Syncytial Virus (RSV) [3]. Daily measurements of pulmonary health markers such as the respiratory rate and tidal volume can provide vital information about the functioning of the lungs [4,5]. Recent evidence suggests that the respiratory rate is an indication of respiratory health: over 20 breaths/min (bpm) indicates unwell, >24 bmp indicates critical illness [4], and >30/min indicates severe disease [6].

Spirometry tests are highly accurate measurements of respiratory health; however, these methods are limited to clinical settings, due to their bulkiness and cost, and require skilled staff to operate [7]. Efforts have been made to leverage spirometry in a home setting; however, they resulted in an underestimation of pulmonary health markers when compared with spirometry performed in clinics [8].

Impedance pneumography, including that from wet electrodes or within wearables, has been proposed to enable the assessment of pulmonary health remotely [9,10]. These devices non-invasively measure the thorax impedance using wet electrodes attached to the chest (Figure 1a), which inject a small amount of current into the patient’s body and measures the resulting impedance based on the concept that impedance increases during inhalation, as the chest expands, and decreases during exhalation, as air leaves the lung. This method has shown a good correlation (higher than 0.9) of breathing with impedance; however, it still lacks adoption in medical practice and home follow-up treatment [9,11]. This is due to its limitations, which include a high variability resulting from a sensitivity to electrode placement and significant skin–electrode contact impedance [12].

In our prior work [13], we designed a novel approach to monitor respiration by measuring changes in impedance from the back of the thigh using the bioimpedance technique (Figure 1b). To summarize, four dry titanium electrodes were integrated into a toilet seat, and we measured the time-dependent impedance at a single frequency. The integration of electrodes into a seat provided a repeatable electrode placement and a large area of skin contact. The relationship between thigh–thigh impedance and respiration was investigated empirically by comparing the thigh–thigh bioimpedance measurements to spirometer measurements. A measurable impedance change was obtained with respect to shallow, normal, deep, and maximal breathing. We also explored the mechanistic principle that leads to the detection of respiration from this electrode location computationally using finite element modeling (FEM). The FEM results indicated that geometrical changes related to abdominal compression led to more significant variations in the thigh–thigh impedance magnitude than the changes in lung volume did. We showed the feasibility of using thigh–thigh bioimpedance measurements for measuring the respiration rate and demonstrated a linear relationship between thigh–thigh bioimpedance and lung volume. The slope of the linear relationship was patient-specific and only measured on a single day. A practical implementation of bioimpedance for monitoring daily changes in respiration requires a quantitative understanding of the variability, and this has not yet been explored.

The goal of this work is to evaluate the feasibility of tracking daily changes in respiration using measurements of thorax and thigh–thigh impedance. To achieve this, we quantify the variability in impedence measurements on healthy subjects, with the idea that the variability that is inherent to the measurements places a lower limit on the minimum detectable physiologic change that the system could detect if eventually used for in-home monitoring. This investigation has been carried out across five healthy subjects by conducting within-session and day-to-day impedance measurements for three consecutive days using dry electrodes (thigh) and wet electrodes (thorax) across the same healthy subject, along with simultaneous spirometer measurements for validation. Additionally, factors such as the frequency, skin hydration, and effect of food and water before and after bioimpedance measurements have been investigated and quantified.

## 2. Materials and Methods

### 2.1. Bioimpedance Measurements

Thigh–thigh bioimpedance measurements were conducted using a MAX30001 Analog Frond End (Maxim Integrated, San Jose, CA, USA) at 80 kHz and 48 μA in a 4-electrode configuration using dry titanium electrodes integrated into a toilet seat (Figure 2). Respiration rate and volumetric flow were measured simultaneously with a spirometer (MTL1000L, AD Instruments, Colorado Springs, CO, USA). The subject was asked to sit upright with their hands on their lap and instructed to perform a 2 min sequence of respiratory tasks: shallow–normal–deep–maximal inhale/exhale. For comparison, bioimpedance measurements were performed using wet electrodes across the thorax, again synchronized with the spirometer. The details regarding the signal processing steps involved in analyzing the spirometer volume signal to the impedance signal can be found in our previous work [13]. The signal processing steps are also shown in Figure 3. To summarize, the spirometer flow signal was obtained at 1000 Hz and integrated to obtain the lung volume. Simultaneous to the spirometer measurements, impedance measurements were acquired at 64 Hz. The acquired spirometer volume and impedance signal were resampled to 100 Hz for consistent comparison. Three seconds of data were removed from the start and end for both the spirometer volume and raw impedance recordings, as it takes a few seconds to settle and scale. The baseline drift of the raw impedance signal was removed using polynominal detrending, as we are interested in the relative amplitude (peak–peak) of the signals. Thereafter, a third-order Savitzky–Golay filter was used to smoothen the raw impedance signal. The peak-to-peak value of the spirometer volume signal (ground-truth tidal volume) was correlated to the peak-peak value of the impedance signal (aligned using cross-correlation) (Figure 3). The relationship between the respiratory volume measured by the spirometer volume signal and the impedance signal was computed by using linear regression with y-intercept forced to zero (Figure 4).

### 2.2. Bioimpedance Variability Experiments

**Within-session variability**: To evaluate the day-to-day variability, four bioimpedance measurements and simultaneous spirometer measurements were obtained with each of the dry (thigh) and wet (thorax) electrodes on the same day and within the same session. The subject was asked to stand and sit between each replication of the thigh–thigh impedance measurements, and the placement of the wet electrodes across the chest was not altered between the measurements. Signal processing and mapping between the peak–peak impedance signal and spirometric volume signal were performed. The slopes obtained after performing linear regression were compared across the four replicates to explore the within-session variability between the bioimpedance measurements across the thigh–thigh and thorax. The mean and standard deviation (SD) of the slopes across four replicates were calculated. Further, the variability between replicates was quantified by calculating the coefficient of variation (COV). The COV was obtained by dividing the SD by the mean.

**Day-to-day variability:** To understand the variability in the bioimpedance measurements across different days, bioimpedance measurements using the dry (thigh) and wet (thorax) electrodes were repeated for three consecutive days. Each day, four impedance measurements, along with the simultaneous spirometer volume signal, were obtained using the dry (thigh) and wet (thorax) electrodes. The measurements were conducted at the same time of the day for three consecutive days. Mapping between the peak–peak thigh–thigh impedance signal and spirometric volume signal was performed, and the mean of the linear regression slopes across four measurements obtained for each day was computed. Further, the COV (SD/mean) was calculated to quantify the variability between Day 1, Day 2, and Day 3. A similar procedure was followed to compare the thorax measurements obtained across the three days.

**Human subject testing:** Within-session and day-to-day bioimpedance variability experiments were performed across 5 human subjects (28.0 ± 3.0 years; BMI: 26.8 ± 6.6), under informed consent, in a protocol approved by the Rochester Institute of Technology Institutional Review Board for Protection of Human Subjects. The correlation slopes were obtained for each day. A paired *t*-test was used to compute if the differences between the slopes obtained for thigh–thigh impedance and thorax impedance each day were statistically significant (*p* ≤ 0.05). The results obtained for within-session variability, day-to-day variability, and inter-subject variability are shown in Section 3.2, Section 3.3 and Section 3.4, respectively.

### 2.3. Factors Affecting the Sensitivity of Bioimpedance Measurements

**Excitation frequency:** To evaluate the effect of the excitation frequency on bioimpedance, thigh–thigh and thorax impedance measurements were captured across 1 subject for at following frequencies: 1 kHz, 2 kHz, 4 kHz, 8 kHz, 16 kHz, 40 kHz, 80 kHz, and 128 kHz. The absolute and relative impedance magnitude were computed to understand the effect of the frequency in the acquisition of the respiration signal.

**Skin hydration:** Skin hydration has typically been considered a crucial factor leading to the variability in the absolute value of bioimpedance measurements that are captured using dry electrodes [12]. The goal of this work was to measure the respiration signal which corresponds to the peak–peak value (relative amplitude) of the impedance signal. Therefore, the effect of skin hydration was investigated by carrying out thigh–thigh impedance measurements using dry electrodes integrated into a toilet seat. Following that, a gel (Spectra 360, Parker Laboratories) was applied to the dry electrodes, and the experiments were repeated. Four replicates were obtained for each experiment across the same subject. The subject was asked to stand and sit between the measurements. The experiments were performed for three consecutive days. The mean and standard deviation of the replicates obtained for the thigh–thigh slope were computed for both dry and gel electrodes. A paired *t*-test was used to compute if the differences between the slopes obtained for thigh–thigh impedance and thorax impedance were statistically significant (*p* ≤ 0.05).

**The effect of the abdomen and bladder:** In our prior work [13], computational modeling revealed that the thigh–thigh current path predominantly passes through the pelvis and thigh area. Therefore, the impact of the abdomen on bioimpedance measurements was examined by performing thigh–thigh and thorax impedance experiments across a subject before and after breakfast. To evaluate the effect of the bladder, impedance experiments were performed before drinking water, after drinking water, and after urination. A similar analysis to that of the skin hydration experiments was performed to evaluate the statistically significant differences between the slopes obtained for the “before food” and “after food”, “before water”, “after water”, and “after urination” experiments.

## 3. Results

### 3.1. Bioimpedance Measurements

For each subject (12 replicates across three days), the peak–peak bioimpedance measurements were found to be highly correlated (0.94 ± 0.03 for dry electrodes across the thigh; 0.92 ± 0.07 for wet electrodes across the thorax) with the peak–peak spirometer tidal volume. The high correlation of wet electrodes across the thorax is consistent with the literature [14]. The empirical results for the thigh–thigh measurements show a high correlation and linear relationship between the thigh–thigh impedance and respiration.

### 3.2. Within-Session Variability

Figure 5 depicts the relationship between the tidal volume and impedance for a single subject (subject 2). Each data series and resulting regression is a replicate experiment, and slopes for all subjects are shown in Appendix A Appendix A. As an example, we summarize here the results for a single subject (2) and day (2). The COV beween replicates for subject 2 on Day 2 is 7% for thigh–thigh and 4% for thorax. The COVs between replicates for all subjects on Day 2 were 16%, 7%, 16%, 8%, and 9% for thigh–thigh and 23%, 4%, 17%, 34%, and 2% for thorax. The within-session variability was higher for thigh–thigh than for the thorax for two out of five subjects on this day. The within-session COV was typically lower for thigh–thigh impedance (an average across all subjects for all days of 11.2%) compared to the thorax (15.6%).

Most subjects’ tidal volume varied between 0.5 and 4 L. The peak–peak thigh–thigh impedance (0.02 Ω to 0.23 Ω) was an order of magnitude lower than the peak–peak thorax impedance (0.1 Ω to 2.4 Ω) over this range. Both the thigh and thorax locations were suitable for detecting impedance changes over this range. This shows the high sensitivity of both locations for picking up shallow breathing cycles.

### 3.3. Day-to-Day Variability

Figure 6 shows the slopes of the linear regression obtained across three days for subject 2. The day-to-day COVs are 22% for thigh–thigh and 27% for the thorax. The data depicting the day-to-day variation for this and all other subjects are shown in Figure 7. For the measurements across three days, the day-to-day COVs are 9%, 22% 7%, 24%, and 7% for thigh–thigh and 29%, 27%, 38%, 94%, and 14% for thorax for subjects 1, 2, 3, 4, and 5, respectively. This indicates that the day-to-day variation in thigh–thigh is significantly lower than the thorax COV, which is likely due to the challenges with a consistent placement of the thorax electrodes [15,16]. The high day-to-day variability in the thorax measurements obtained in this work is consistent with that of Grenvik et al. [17], where changes of 53% and 64% were observed in the slopes depicting the changes in chest impedance to that of changes in the lung volume of two subjects on two different days.

### 3.4. Inter-Subject Variability

A wide range of correlation slopes for the thigh (17.4 to 74.3) and thorax (0.6 to 13.5) were obtained across five subjects (Figure 7). The correlation slope between the peak–peak impedance and spirometric volume signal obtained across five subjects was significantly different for both the thigh and thorax locations (Figure 7). This indicates that a subject-specific calibration is needed. This is consistent with the literature [18,19]. For subjects 1 and 3, the correlation slope was not significantly different for Day 1, Day 2, and Day 3 for thigh–thigh; however, it was significantly different for the thorax (*p* < 0.05). For subjects 2 and 4, the correlation slope was significantly different (*p* < 0.05) for Day 1, Day 2, and Day 3 for both thigh–thigh and the thorax. For subject 5, the correlation slope obtained on different days was not significantly different for either thigh–thigh or the thorax. Figure 7 shows that four out of five subjects had significantly different slopes for the thorax location using wet electrodes, and two out of five did for the thigh location. This indicates that the thigh location has a relatively lower variability.

### 3.5. Effect of Excitation Frequency

With the increase in frequency, the absolute value of thorax and thigh impedance was found to decrease. This is in agreement with the literature [20,21]. However, the goal of this work was to evaluate the effect of the frequency on the relative amplitude, which is the detection of the change in impedance due to respiration. See Figure 8.

Figure 9 shows that the value of the peak-to-peak impedance for both the thorax and thigh was not affected by the excitation frequency ranging from 1 kHz to 128 kHz. For thigh–thigh, the largest change in peak–peak impedance was obtained at 80 kHz. Therefore, for this work, we chose 80 kHz as the excitation frequency, which is in agreement with the literature [22].

### 3.6. Effect of Hydration

A paired *t*-test of the thigh–thigh slope obtained for dry and dry + gel electrodes for each day resulted in *p*-values of 0.058 (Day 1), 0.35 (Day 2), and 0.48 (Day 3). Since the *p*-values were greater than 0.05, this indicates that there were no significant differences between the dry and gel electrodes. These results indicate that the gel on electrodes is not a significant factor and does not affect the reproducibility of the bioimpedance measurements as a function of respiratory volume.

### 3.7. Effect of Food and Water

The correlation slope captured across the thigh–thigh and thorax resulted in a 16% percentage change in the relative value of impedance before and after water. There were no significant changes in the correlation slope across thigh–thigh and thorax after urination. This indicates that impedance measurements on a daily basis might be affected by the consumption of water or the amount of water in the body. This is in agreement with Russo et al., where the body impedance increased by 6–17 Ohms in comparison with values in the fasting state [23]. The correlation slopes before and after food resulted in a 36% and 8% percentage change for thigh–thigh and the thorax, respectively. The paired *t*-test depicts that the slopes were significantly different for the thigh–thigh but not for thorax impedance measurements. This indicates that thigh–thigh impedance measurements might be affected by the state of the abdomen.

## 4. Discussion

In this work, we investigated the practicality of tracking daily changes in respiration by measuring the impedance across the thighs using dry electrodes integrated into a toilet seat. The day-to-day variability and within-session variability in bioimpedance measurements were studied to understand the feasibility of the technique for detecting an actual change in the respiratory rate and tidal volume. The thigh–thigh results were compared to the impedance measurements across the thorax, which is a more typical location [24,25]. The impedance measurements were validated by capturing simultaneous gold standard spirometry measurements.

Our study showed that the absolute magnitude of bioimpedance was an order of magnitude lower for the thigh–thigh that for the thorax. This could be due to a shift in the measurement location from the thorax to the lower body location (thigh). The within-session COV was typically lower for thigh–thigh impedance (an average across all subjects for all days of 11.2%) compared to the thorax (15.6%). This low COV for thigh–thigh was achieved despite the subjects standing and sitting between each replicate, whereas the electrodes were not moved between within-session replicates for the thorax. Data across all subjects for all days are shown in Appendix A Appendix A. These show that the integration of electrodes into a seat provided a repeatable electrode placement. The within-session variability for thigh–thigh was small enough that a single daily measurement is likely sufficient for home monitoring and monitoring longer-term trends.

For the representative subject (2), the day-to-day variability was higher (27%, Figure 6) than the within-session variability (4%, Figure 5) for the wet electrodes, as well as for the thigh–thigh electrodes (22% vs. 7%). Figure 7 shows that this inference is also valid for most subjects, with the day-to-day variability (thigh–thigh: 9%, 22%, 7%, 24%, and 7%; thorax: 29%, 27%, 38%, 94%, and 14%) being higher than that of within-session (thigh–thigh: 16%, 7%, 16%, 8%, and 9%; thorax: 23%, 4%, 17%, 34%, and 2%). The large day-to-day variability for thorax electrodes is likely due to the differences in the positioning of the electrodes on consecutive days [26]. The thigh–thigh measurements had lower variation than the thorax measurements, which demonstrates the technique’s potential to detect actual physiological changes in tidal volume.

The variability resulting from both within-session and day-to-day variability for thigh–thigh was patient-specific (17.4 to 74.3 across the five subjects; Figure 7a). This might be due to the differences in the BMIs of the subjects and the difference in their body mass compositions, especially across the torso and thigh area. It is clear that patient-specific data will be developed when used, but this seems feasible, as the point of the device is to look for trends in the data rather than to measure absolute values. The range of thorax slopes varied similarly to the thorax measurements (0.6 to 13.5; Figure 7b). We believe that the wide range of correlation slopes for the thorax, especially for subject 3 and subject 4 is because of the inconsistent positioning of wet electrodes across the chest.

We looked into factors such as the excitation frequency, skin hydration, food, and water to understand if they contribute significantly to within-session or day-to-day variability in bioimpedance measurements. Excitation frequencies ranging from 1 kHz to 128 kHz did not affect the peak-to-peak impedance (relative value) of the bioimpedance. Figure 10 further indicates that the day-to-day variability in bioimpedance measurements obtained with dry electrodes across the thighs was not affected by skin hydration. Moreover, the presence of a gel on the dry electrodes did not significantly affect the thigh–thigh slope obtained with dry electrodes, as we were measuring the change in impedance due to respiration, which corresponds to the relative amplitude of the impedance and not the absolute value of impedance. The correlation slope for thigh–thigh bioimpedance measurements was found to be significantly different after consuming food and water, which indicates that the bioimpedance measurements might be affected by the contents in the abdomen and bladder. See Figure 11.

We showed in this and earlier works that the thigh–thigh bioimpedance depicts a high correlation to the respiration rate and volume signal. The five subjects in this study were sufficient to understand the day-to-day variability. It indicates that the day-to-day variability in the relationship between the impedance and volume might limit the accuracy and sensitivity of the measured respiratory tidal volume. If an actual physiological change is larger than the day-to-day variation that we measured here, this method would be able to detect that change in physiological state with a single measurement. Although this threshold is subject-specific, the day-to-day variability was small (7–24%) for thigh–thigh and only statistically significantly different from day to day for two subjects, one day each. For the thorax, the day-to-day COV was larger (14–94%) and had substantially more days that were significantly different from one another (four of the five subjects). This indicates that the thigh–thigh location has the potential to overcome the challenges faced by the thorax location. In this work, experiments were carried out in a controlled environment for three consecutive days. If performed on a regular basis for one person, it seems quite feasible that a patient-specific calibration and understanding of day-to-day variation could be developed by the algorithm during in-home use.

## 5. Conclusions

This work shows that the thigh–thigh bioimpedance exhibits a high correlation with the respiration rate and thus can be effectively used to measure the respiratory rate and breathing patterns. The linear relationship between the thigh–thigh impedance and lung volume provides an approach that can be used to estimate the tidal volume.

The measurement of the respiration rate from a thigh–thigh electrode position is straightforward. The relationship between the thigh–thigh impedance and tidal volume is patient-specific and varies within a session and from day to day. The results of the day-to-day variability experiments across five healthy subjects suggest that the day-to-day variations in bioimpedance measurements across the thighs are smaller than similar measurements from the thorax location. Thus, the thigh location compared to the thorax location has the potential to better detect changes in tidal volume resulting from disease onset or recovery. The results suggest that the excitation frequency and skin hydration do not play a significant role; however, the consumption of food and water can lead to variations. As is the case for many health metrics that are measured daily (weight, blood pressure), it is important that the measurement is taken at the same point in a daily routine.

## Figures and Tables

**Figure 1 sensors-24-04612-f001:**
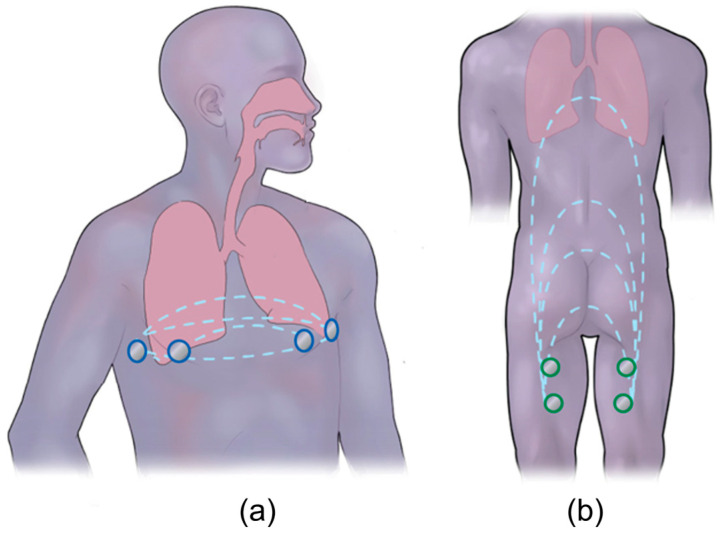
Overview of bioimpedance technique: (**a**) traditional way of performing the bioimpedance measurements across the thorax, where two outer electrodes inject a current and the two inner sense the voltage (blue), which further provides impedance as a function of breathing. (**b**) A novel approach of respiration detection from a non-standard location, electrodes on the back of the thigh (green), using bioimpedance is presented, which can aid with in-home physiological monitoring (modified from [13]).

**Figure 2 sensors-24-04612-f002:**
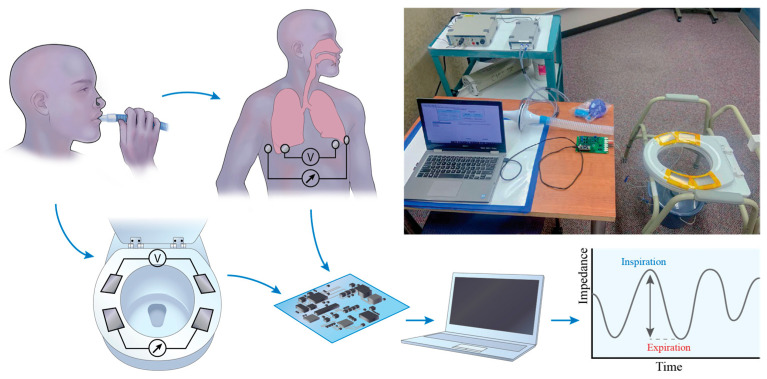
Experimental setup for simultaneous spirometer measurements for measuring respiratory volume along with bioimpedance measurements from the back of the thigh (dry electrodes on the seat) and wet electrodes across the thorax (reproduced from [13]).

**Figure 3 sensors-24-04612-f003:**
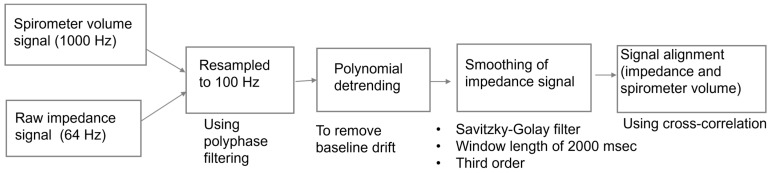
Steps followed for preprocessing and aligning the spirometer volume signal to the impedance signal.

**Figure 4 sensors-24-04612-f004:**
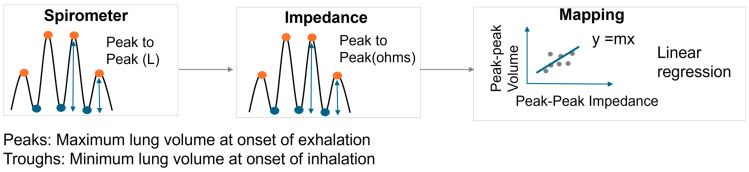
Mapping between peak–peak impedance signal and spirometric volume signal.

**Figure 5 sensors-24-04612-f005:**
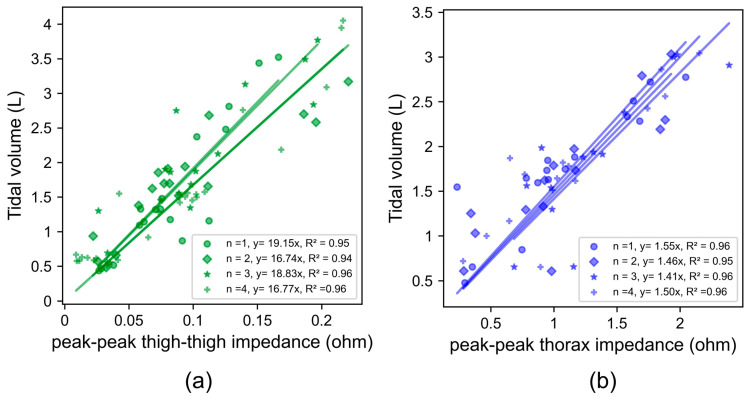
Least squares linear regression between the peak–peak impedance across thigh–thigh and the ground-truth tidal volume, measured using a spirometer for (**a**) dry electrodes across thigh–thigh and (**b**) wet electrodes across the thorax. Slopes obtained within the same session for four back-to-back measurements are shown as different shapes: *n* = 1 (dot), *n* = 2 (diamond), *n* = 3 (star), and *n* = 4 (plus).

**Figure 6 sensors-24-04612-f006:**
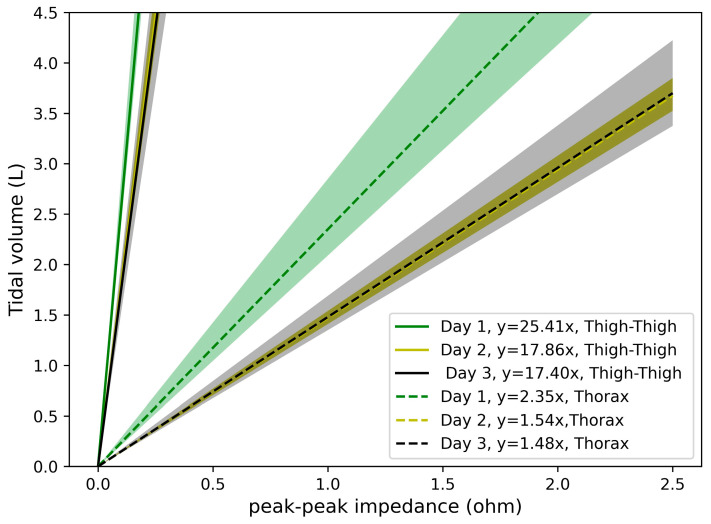
Least squares linear regression between the peak–peak impedance and the ground-truth tidal volume, measured using a spirometer. Shown for one subject for dry electrodes across thigh–thigh (solid lines) and wet electrodes across the thorax (dashed lines). Mean linear regression and one standard deviation is shown: Day 1 (green), Day 2 (yellow), and Day 3 (black).

**Figure 7 sensors-24-04612-f007:**
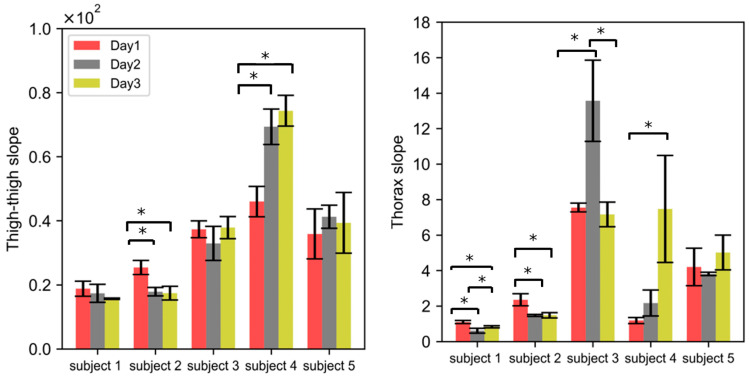
Evaluation of the variability in the impedance measurements across different subjects. Each bar graphs shows the mean of the four replicates within a single session, and error bars denote the standard deviation (aka within-session variability). The means of the correlation slopes obtained on Day 1, Day 2, and Day 3 are represented by different colors. (*) denotes the statistical significance of *p* < 0.05, obtained with a paired *t*-test.

**Figure 8 sensors-24-04612-f008:**
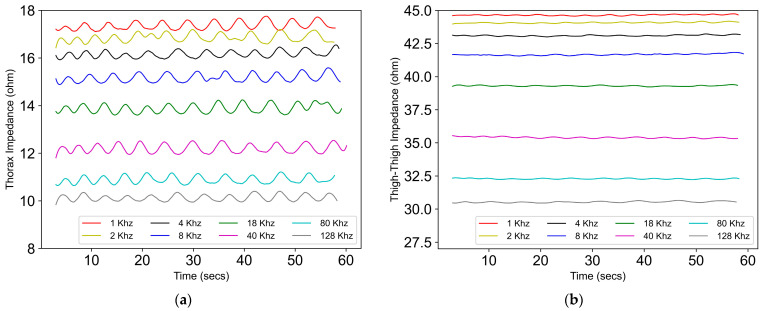
The absolute value of the impedance signal across different excitation frequencies, ranging from 1 kHz to 128 kHz across (**a**) the thorax and (**b**) thigh–thigh.

**Figure 9 sensors-24-04612-f009:**
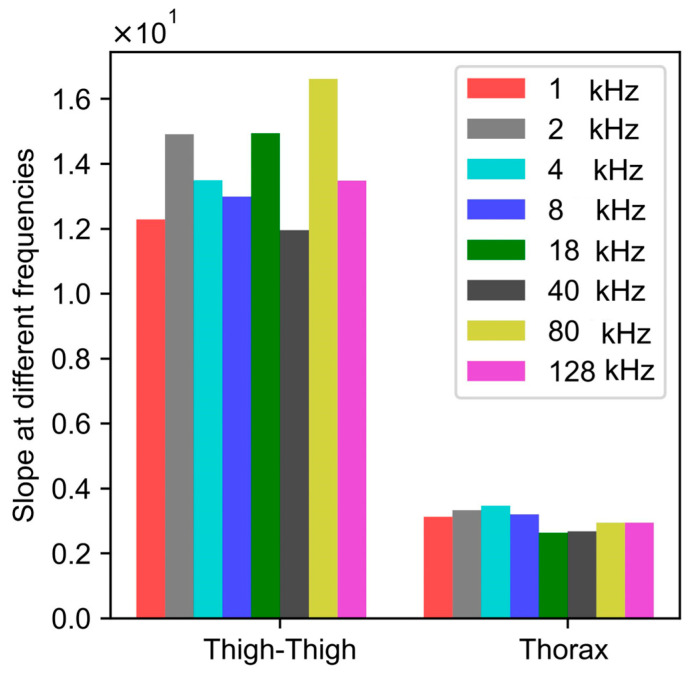
Correlation slopes at different excitation frequencies, obtained by performing mapping between peak–peak change in impedance and the spirometric volume signal. The peak–peak value of impedance depicts the relative change in the value of the impedance.

**Figure 10 sensors-24-04612-f010:**
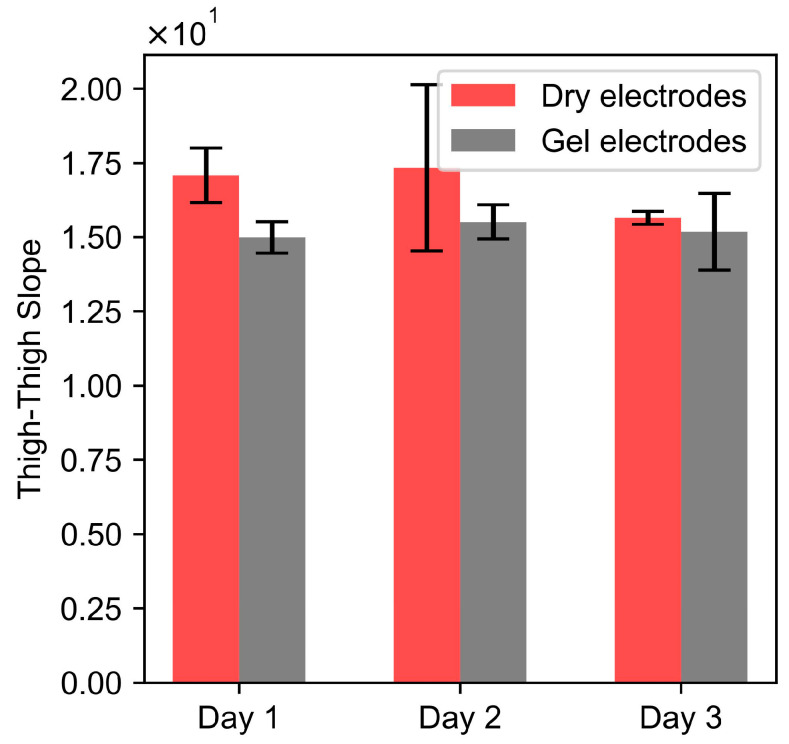
Rectangular bars show the mean of the replicates for the thigh–thigh slope, computed for dry and dry + gel electrodes for three consecutive days. *n* = 4 for each category, and the height of the error bars denotes the standard deviation.

**Figure 11 sensors-24-04612-f011:**
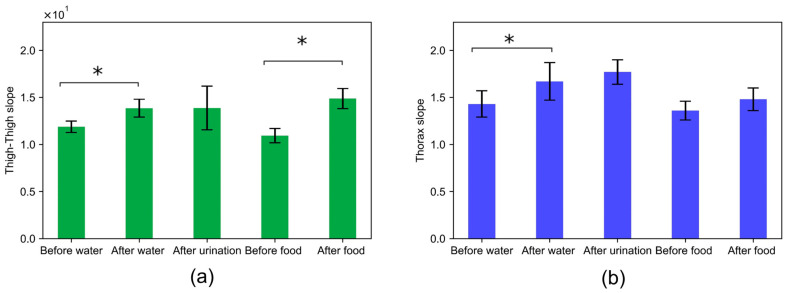
Evaluation of the variability in bioimpedance measurements before and after food and water consumption and after urination across (**a**) thigh–thigh and (**b**) the thorax. Rectangular bars show the mean for a slope computed for impedance measurements conducted before water, after water, after urination before food, and after food. *n* = 4 for each category, and the height of the error bars denotes the standard deviation. (*) denotes the statistical significance of *p* < 0.05, obtained with a paired *t*-test.

## Data Availability

The original contributions presented in the study are included in the article, further inquiries can be directed to the corresponding author.

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
