# Peer review of "Day-to-Day Variability in Measurements of Respiration Using Bioimpedance from a Non-Standard Location"

_sensors, 2024, doi:10.3390/s24144612_

Round 1

Reviewer 1 Report

Comments and Suggestions for Authors

This paper presents a method of measuring the respiration of a patient using thigh-thigh bioimpedance. This work depicts a high correlation to the respiratory rate and thus can be effectively used to measure respiratory rate and breathing patterns. This work is a good contribution to the related field. The authors are requested to revise the paper before further consideration of acceptance.  Here are some comments:

(a) It seems this work is an extension of the authors' previous work presented in reference [13].  and the authors have referred to their work a couple of times in the paper. The authors are requested to present their previous work and the major findings in this paper in the introduction section (Section 1, second paragraph). This will make the paper more complete. Especially, the signal processing steps used in the previous work are worthwhile to mention in this paper. 

(b) The authors are requested to modify the reference of their previous work (i.e. reference [13]). There is no name of the conference, date, location etc. I think this reference should be cited as "K. Goyal, D. A. Borkholder and S. W. Day, "Unobtrusive In-Home Respiration Monitoring Using a Toilet Seat," 2022 IEEE-EMBS International Conference on Biomedical and Health Informatics (BHI), Ioannina, Greece, 2022, pp. 01-05, doi: 10.1109/BHI56158.2022.992693"

(b) Similarly, the reference [26[ is not complete. I think it will be written as "Wang HB, Yen CW, Liang JT, Wang Q, Liu GZ, Song R. A robust electrode configuration for bioimpedance measurement of respiration. J Healthc Eng. 2014;5(3):313-27. doi: 10.1260/2040-2295.5.3.313. PMID: 25193370."

(c) The authors are requested to improve the resolution of Fig. 2. This figure is hardly visible. 

(d) This paper requires a detailed explanation of the signal processing steps presented in Fig. 3.  This is very important for the reader to understand the signal processing steps followed by the authors to implement the project. 

Reviewer 2 Report

Comments and Suggestions for Authors

The authors presented an attempt to demonstrate the correlation between the peak value of lung capacity measured with a spirometer and the peak value of the impedance modulus measured at one frequency between the patient's thighs. They compared the obtained results with analogous studies for electrodes placed on the chest.

The article is interesting, but I think it lacks justification for why changes in impedance between the thighs should reflect changes in lung capacity during breathing. This is not intuitively obvious. The thighs are located quite far from the lungs, why would such rapid changes in breathing have a direct impact on the change in impedance between the thighs? Impedance changes measured between the thighs have a much smaller amplitude than those measured on the chest (Figure 8). What is the error in the impedance measurement under these conditions? Maybe another signal induces the formation of this signal and the dry electrodes act as antennas?

It is not clear how the spirometer results are converted into an electrical signal (left side of Figure 3). Please describe it in more detail. Isn't there a connection between this circuit and the thigh impedance measurement by electromagnetic induction? Was electrode shielding used to avoid interference (something like a Faraday cage)? Why does the presence of gel on the electrodes not matter (lines 266-269)? It may be worth conducting an impedance test between the thighs in one person and measuring the impedance with a spirometer in the other person sitting next to him. Would similar impedance changes be observed?

I think these issues need to be resolved.

Minor remarks.

1. There is no exact affiliation of the authors (first page, under the title), only the name of the department is not enough.

2. In Figure 9, the unit of frequency is kHz. Other entries are incorrect.

3. Some data in References is incomplete, for example [13]. Please check and complete.

Comments on the Quality of English Language

English is understandable.

Reviewer 3 Report

Comments and Suggestions for Authors

Line 52:

For Figure 1(b), there is no evidence that the current between two electrodes can reach up to the back which is lower parts of longs. Usually, the current will take the shortest path between two electrodes. Unless the authors measure and proof the paths of the electrodes using other measurement devices, Figure 1 b) cannot be accepted. It might be too much overstated without the obvious facts.

Line 84:

The authors need to clarify why they choose 80KHz single frequency.

(As an example, the gold standard of most of the single frequency in bioelectrical impedance measurements is 50Khz)

Also, The authors need to mention regarding 48uA current. Is that enough or have you tried other setting? Under 100uA from the current pump  is also usually low side.

Line 85:

The poses that the patient sitting on the toilet seat also need to be addressed and canonized for the consistence of the experiments. The position of arms and hands need to be addressed.

Line 135:

Five human subjects are too small to make general conclusions or arguments.

Line 145:

The excitation frequency should be between 40-100Khz and low frequency will not much be needed for the single frequency point of view.

Line 157:

The amount of gel for the interface between dry electrodes and subject skin is hard to control. Also, the pressure between the electrodes (dry or wet) can be very crucial for the measurement results. Have you considered the pressures or weight of the patients?

Line 221:

Yellow dashed line for day 2 is hard to visualized in the graph.

Line 251:

What are the authors intentions from Figure 8?

Showing the results of the different frequencies does not helpful in this manuscript.

Line 259:

Low frequencies (below 50Khz) usually involved in ECF and high frequencies above 50khz take care of ICF. Are these results strange? Thigh has more muscle which is related to ICF and so on. Why and how these results are like each other for the different frequencies? If we measure all this frequencies, it should be the part of the cole-cole curve (Almost the right half of the cole-cole curve)

Line 353:

What kind of day-to-day variation are we referring to? What algorithms can you draw form the data you collect? How can you utilize the collected data? Do you need other biomarkers to compare these data or can the data be used stand alone?

Round 2

Reviewer 1 Report

Comments and Suggestions for Authors

The authors have implemented all the comments and suggestions. The paper is more informative than the previous version. The quality of the paper is much better now—many thanks to the authors for their hard work and dedication. 

Author Response

Thank you very much for your help in improving the manuscript.

Reviewer 2 Report

Comments and Suggestions for Authors

The authors improved the manuscript as a result of the reviewers' comments. However, after the authors' explanations, I am still concerned about the impact of a regular heartbeat on the measurement of impedance values (cardiogenic oscillations). Does this affect the result? The heartbeat also compresses the organs in the lower part of the body at a frequency similar to that of breathing, especially if the patient breathes from the diaphragm. Have the authors considered this and verified it experimentally? This needs to be discussed. This is justified by the fact that an identical measurement system was used to assess the heart rate (Acta Polytechnica Vol. 51 No. 5/2011, Unobtrusive Health Screening on an Intelligent Toilet Seat, T. Schlebusch).

Please resolve whether we are measuring heart function or respiration.

It is also worth citing other authors who used a similar measurement system (with a toilet seat) to measure bioimpedance, which the authors did not do in the manuscript.

Author Response

Comment 1: The authors improved the manuscript as a result of the reviewers' comments.

Response 1: Thank you.  We appreciate your comments, which helped improve the manuscript.

Comment 2: However, after the authors' explanations, I am still concerned about the impact of a regular heartbeat on the measurement of impedance values (cardiogenic oscillations). Does this affect the result? The heartbeat also compresses the organs in the lower part of the body at a frequency similar to that of breathing, especially if the patient breathes from the diaphragm. Have the authors considered this and verified it experimentally? This needs to be discussed. This is justified by the fact that an identical measurement system was used to assess the heart rate (Acta Polytechnica Vol. 51 No. 5/2011, Unobtrusive Health Screening on an Intelligent Toilet Seat, T. Schlebusch). Please resolve whether we are measuring heart function or respiration.

Response 2: We are certain that we are measuring respiration rather than cardiac cycle.  This is very clear due to at least two reasons: 1) the excellent linear proportionality between the actual measured tidal volume from a spirometer to the peak-to-peak impedance, which has an intercept very near 0 (impedance does not change when the subject holds their breath, even though the heart continues to beat).  Secondly, during our test protocol, the individual breathes shallow, normal, and deep (line 98), which results in a >4 fold variation in tidal volume and impedance.  There is no reason to think that cardiac variability would have this large a range in magnitude (Fig 5a).

Our understanding of Schlebusch is that heart rate was detected by ECG (voltage potential, “recorded ECG signal is dependent on the strength of the potential difference between the electrode positions. From two opposite positions on the chest, positions A and B in Figure 3, a higher amplitude can be measured than between positions C and D, as we have for the ECG measurement on a toilet seat.”), whereas the impedance measurements were used “… for determining body composition”

Comment 3: It is also worth citing other authors who used a similar measurement system (with a toilet seat) to measure bioimpedance, which the authors did not do in the manuscript.

Response 3: In our previous published work [reference 13] we already cited in the discussion section and compared our work to others work including the reference suggested by the reviewer. Those studies using the toilet seat setup have been focused on measuring the absolute values of impedance, whereas we are interested in detection of the change in impedance due to respiration. 

If the reviewer feels that more of this should be made explicit in the manuscript revision, please let us know exactly what points in our response above.